# Effect of Intragastric Botulinum Type A Injection Combined with a Low-Calorie High-Protein Diet in Adults with Overweight or Obesity

**DOI:** 10.3390/jcm11123325

**Published:** 2022-06-10

**Authors:** Po-Ke Hsu, Chia-Lin Wu, Yu-Hsuan Yang, James Cheng-Chung Wei

**Affiliations:** 1Institute of Medicine, Chung Shan Medical University, Taichung 40201, Taiwan; berger1912@gmail.com; 2Department of Weight Control Center, Sun Saint Clinic, Zhubei City 302052, Taiwan; shtstar@hotmail.com; 3Department of Gastroenterology, Changhua Christian Hospital, Changhua City 500209, Taiwan; 4Department of Nephrology, Changhua Christian Hospital, Changhua City 500209, Taiwan; 143843@cch.org.tw; 5Department of Allergy, Immunology & Rheumatology, Chung Shan Medical University Hospital, Taichung 40201, Taiwan; 6Graduate Institute of Integrated Medicine, China Medical University, Taichung 40201, Taiwan

**Keywords:** obesity, overweight, intragastric botulinum injection type A injection, obesity therapy, endoscopy

## Abstract

(1) Background/aims: Intragastric botulinum toxin A injection (IGBI) combined with diet control is a new and effective weight loss method for grade 2 obese patients. However, the application of IGIB on overweight or obese adults still needs further research to confirm its efficacy. (2) Methods: We retrospectively collected medical data from 1 July 2021 to 1 January 2022 from a total of 71 patients without diabetes who participated in the bariatric clinic with a body mass index (BMI) > 25 kg/m^2^. Forty-nine participants opted for intragastric botulinum injection (IGBI) using 300 units of botulinum injected into the antrum, body, and fundus, followed with a low-calorie high-protein diet course. Another 22 people participated only in the low-calorie high-protein diet course as a placebo group. This study analyzes the weight loss percentage of the two groups. Adverse events after IGBI are also reported in a safety assessment. (3) Results: In terms of the characteristics of the two groups, the mean BMI was 29.3 kg/m^2^ in the IGBI group and 28.0 kg/m in the placebo group (*p* = 0.63 without significant difference). Comparing the percent weight loss from baseline in the two groups after 12 weeks, the IGBI group lost 11.5% of their body weight and the placebo group lost 1.8%. In terms of group analysis, the percentages of participants with a weight reduction of at least 5% for the IGBI and placebo groups were 95% and 4%, respectively. For weight reduction of at least 10%, these values for the IGBI and placebo groups were 63% and 4%, respectively. In terms of adverse events after IGBI for 12 weeks, 12 participants (24.4%) had constipation, which was the main side effect. No serious adverse events were observed during the study period. (4) Conclusion: The combination of a low-calorie high-protein diet and IGBI is an effective and safe procedure in overweight or obese adults for weight reduction, but further larger studies are needed.

## 1. Introduction

According to the World Health Organization (WHO), approximately 1.9 billion adults (over the age of 18) were overweight and more than 600 million were obese in 2014 [1]. By 2030, the number of overweight people is expected to exceed 2.16 billion, and the number of obese people will exceed 1.12 billion [2]. For adults, the WHO defines overweight as body mass index (BMI) greater than or equal to 25 kg/m^2^, and obesity as BMI greater than or equal to 30 kg/m^2^ [3]. In recent years, obesity trends have increased, raising the likelihood of chronic diseases such as cardiovascular disease, non-alcoholic fatty liver disease, stroke, diabetes, hypertension, and hyperlipidemia, and even increasing mortality and morbidity rates in persons with coronavirus disease 2019 (COVID-19) [4,5,6,7]. In the American College of Cardiology/American Heart Association obesity guidelines, all patients recommended for weight loss should be offered or referred for a comprehensive lifestyle intervention [8]. Comprehensive lifestyle intervention with individualized calorie-restricted diet such as a dietary intake of 1200–1500 kcal/d for women and 1500–1800 kcal/d for men is recommended. Dietary control, such as a low-calorie, high-protein diet, is a good choice for weight loss [9]. Medical treatment methods include psychotherapy, and behavioral therapy, but there are still limited weight loss effects in some groups [10,11]. Bariatric surgery—including invasive procedures such as gastric banding, bypass operation, and sleeve gastrectomy—is effective but still has associated irreversible complications [12,13]. In animal studies, intragastric botulinum toxin A injection (IGBI) can delay gastric emptying and achieve weight loss goals by controlling appetite [14,15], prolonging gastric emptying time to produce early satiety, and reducing food intake to achieve the effect of weight loss. However, current real-world data are insufficient to show that IGBI is an effective method for sustained weight loss in obese patients [16]. The aim of this study was to assess the efficacy of IGBI for the treatment of overweight or obese adults using clinical real-world data.

## 2. Materials and Methods

We retrospectively collected data from 1 July 2021 to 1 January 2022 of 76 patients with obesity or overweight at the weight control center of the Sun Saint clinic who met the following study inclusion criteria: (1) age of 18 or older; (2) had signed the informed consent form. Meanwhile, the exclusion criteria were: (1) diabetes; (2) a history of malignancy; (3) peptic ulcers diagnosed within 180 days; (4) BMI less than 25 kg/m^2^; (5) loss of follow-up. Forty-nine participants opted for IGBI, and 300 units of botulinum toxin were injected into the pylorus, body, and fundus of the stomach combined with a low-calorie high-protein diet course. Another 22 people participated only in the low-calorie high-protein diet course: the placebo group. As mentioned, all participants signed an informed consent form for this study, which was reviewed and approved by the institutional review board of Changhua Christian Hospital (no. 210202).

### 2.1. Endoscopic Procedures of IGBI

First, we specified and signed an informed consent form for each participant who chose IGBI. Additionally, we arranged blood tests and physical examinations to ensure that there were no related risks. If all inspection data were satisfactory, we scheduled a time for IGBI. IGBI was performed by qualified and licensed specialists certified by the Digestive Endoscopy Society of Taiwan. All participants underwent esophagogastroduodenoscopy under fentanyl and propofol sedation. IGBI was not performed if gastric ulcers, tumors, or food residue were found during gastroscopy. During the procedure, we mixed 30 mL of 0.9% normal saline with 300 units of botulinum type A (botox; Allergan, Irvine, CA, USA), used a 25-gauge needle, and injected 10 mL into the antrum, body, and fundus, respectively, through submucosal injection, see Appendix A. After the procedure, we used ultrasound to confirm whether there were ascites or free air to ensure there were no complications such as gastric perforation or bleeding. Participants then returned weekly for physical exam records and dietary recommendations.

Next, we designed the low-calorie high-protein diet control program. At the beginning, we tried to understand the dietary status and habits of the participants and recorded them. When entering the program, we gave a recommended dietary composition comprising total calories, about 40% of which were from carbohydrates, 30% from protein, and finally 30% from fat. Finally, we asked the participants to send back their diet photos through the communication software to calculate the calorie and nutrient composition and gave the participants dietary adjustment suggestions.

### 2.2. Definition and Assessment of Fatty Liver

We used ultrasound B mode to evaluate indicators of fatty liver such as: liver brightness, contrast between the liver and the kidney, and appearance of the intrahepatic vessels. If there was no match, a score of 0 meant no fatty liver. A mild fatty liver was a score of 1, which meant that the liver was bright and the other indicators were normal. Moderate fatty liver equated to a score of 2, which meant liver brightness and contrast between the liver and the kidney. Severe fatty liver equated to 3 points, in which liver echogenicity was significantly enhanced and the portal vein wall imaging was poor [17].

### 2.3. Measures

Body weight was measured in the Sun Saint clinic at baseline and followed by every week at the clinic visits after IGBI. Each participant had to take an 8 h fast in the morning and removed coats and shoes for weight measurement. Participants in both the IGBI and placebo groups used the messaging software (LINE Corporation Ltd., Tokyo, Japan) to send back pictures of their diets, so that dietitians could provide immediate daily dietary advice.

### 2.4. End Points and Assessments

The primary endpoint was the percentage change in body weight from baseline to week 12. The secondary end point was the percentage of people who lost more than 5% and 10% of their body weight at week 12 between the two groups.

### 2.5. Safety Assessments

The safety assessments included analysis of the number of people who presented adverse events for the 12 weeks after the IGBI. We also analyzed how many weeks after IGBI was administered the adverse events occurred for, and the overall percentages of participants with complications.

### 2.6. Statistical Analysis

We used the MedCalc^®^ Statistical Software version 20.106 (MedCalc Software Ltd., Ostend, Belgium, 2022; https://www.medcalc.org, accessed on 1 January 2022) for statistical analysis. First, we evaluated the normal distribution of the quantitative variables, in which data are expressed as n (%) for categorical data and as mean ± standard deviation or median (interquartile range) for continuous data. Sample size was calculated in relation to a body weight change of 9.8 kg in the IGBI group and 1.5 kg in the placebo group at 12 weeks based on our previous data, with a 2-sided type I error rate of 0.05, and 80% power to analyze the treatment effect of the IGBI and placebo groups. Associations were measured by using Pearson correlation coefficients, Fisher exact test, or t-tests, as appropriate.

## 3. Results

We retrospectively collected data from 1 July 2021 to 1 January 2022 from 76 patients with overweight or obesity in the weight control program medical database. Thereafter, four patients were excluded due to loss to follow-up and one was excluded due to having a BMI less than 25 kg/m^2^. Finally, 71 participants met the eligibility criteria for the following analysis (see Figure 1).

The characteristics of the study participants are presented in Table 1. The mean body mass index and weight of the IGBI and placebo groups were 29.3 kg/m^2^, 82.1 kg and 28.0 kg/m^2^, 77.3 kg, respectively, with *p* > 0.05. The number (percentage) of overweight and obese patients was 27 (55%) and 22 (45%) in the IGBI group and 12 (54%) and 10 (46%) in placebo group, respectively, without significant difference *p* = 0.66. The mean totals of cholesterol, triglycerides (TGs) and low-density lipoprotein (LDL) were 164.1, 112.9, and 100.1 mg/dL in the IGBI group and 186.0, 158.6, and 134.2 mg/dL in placebo group, respectively (*p* < 0.05). However, the mean triglycerides and high-density lipoprotein (HDL) values were 112.9 and 43.5 mg/dL in the IGBI group and 158.6 and 51.8 mg/dL in the placebo group, respectively (*p* > 0.05).

### Change in Body Weight as End-Point Assessment

In weeks 4 and 8, the IGBI group had a mean weight reduction of 4.1 and 8.5 percent, respectively, compared to 0.3 and 1.1 percent in the placebo group. The estimated mean weight change from baseline to week 12 was −11.5% for IGBI, as compared to −1.8% for the placebo (estimated treatment difference: −9.7 percentage points; 95% CI, −10.9 to −8.4, *p* < 0.001), as seen in Figure 2. The percentages of IGBI and placebo participants who lost at least 5% of their body weight were 95.9% and 4.5%, respectively. Similarly, the percentages of participants who lost at least 10% were 63.2% and 0%, respectively (*p* < 0.001), as seen in Figure 3 Appendix A showed multivariate logistic regression analysis of factors associated with weight loss more than 5% in IGBI group with age and sex adjusted.

## 4. Safety Assessment

Adverse events were recorded weekly after IGBI for 12 weeks. Time to any adverse event after IGIB was 1 week (median, 95% confidence interval (CI): 1–2). The proportion of all participants with adverse events was 40.8% (20 participants), of which constipation was the most common, accounting for 24.4% (12 participants), followed by nausea and pharyngitis, accounting for 4% each. In addition, in the placebo group, the average time of adverse events was one week, and only two people were constipated. Overall, there were no adverse events serious enough to require hospitalization or emergency department visits, as seen in Table 2.

## 5. Discussion

Botulinum toxin A, a neurotoxic protein produced by bacterium *Clostridium botulinum* and a selective acetylcholine inhibitor, attenuates or blocks smooth and striated muscle contractions by preventing the release of the neurotransmitter acetylcholine from the axon terminals of the neuromuscular junction. It is usually widely used in medical beauty, wrinkle removal, stovepipes, and other fields [18,19,20,21].

The functions of the fundus, body and antrum of the stomach are the grinding and pushing of food for mechanical digestion, and part of the antrum is highly correlated with the time of gastric emptying [22,23]. At the same time, the stomach can also regulate hormones such as gastrin and ghrelin in the body, thereby increasing satiety and the digestion of food [24,25]. For this reason, in recent years, there have been many studies on reducing the digestive function of the stomach through intragastric botulinum injection, thereby achieving the effect of weight loss [26].

In vivo and in vitro studies in 2003 and 2004 explored the effects of botulinum type A on the gut sympathetic nerves [27,28]. In addition, James et al. also reported in an in vitro animal study that botulinum type A effectively reduced the contraction of the antrum muscle [29]. Gui et al. demonstrated that the injection of botulinum toxin A into the antrum was effective in reducing food intake by 37.8% and weight loss by 14% in rats, and the effect persisted for up to 2 months [30].

In 2003, Rollnik et al. documented the first case report of antral injections of botulinum A toxin for the treatment of obesity and showed that, 4 weeks after the injection, weight reduced from 100.6 kg to 91 kg and BMI decreased from 31 kg/m^2^ to 29 kg/m^2^ [31]. Later, a pilot study also showed that botulinum toxin antral injection can delay gastric emptying time and cause a decrease in appetite, thereby allowing obese patients to lose weight [32].

From 2005 to 2013, a total of seven studies investigated IGBI for obesity, including five randomized control trials (RCT) [33,34,35,36,37]. The number of participants in the seven studies ranged from 6 to 30. IGBI doses ranged from 100 units to 500 units and were administered to the antrum; antrum and body; or antrum, body, and fundus of the stomach for a total tracked period from 5 to 24 weeks. In the five RCT groups, the percent of weight loss ranged from 4.9% to 9.0%, indicating that IGBI was effective [38,39].

Compared to our study, IGBI combined with diet control resulted in an 11.5% higher weight loss than the previous RCT study. Compared with the 1.8% weight loss of the placebo group in our study, the IGBI group also had a better weight loss effect. This may be due to the higher doses of botulinum toxin A (300 unites) used in our study, which were injected into the fundus, body, and antrum submucosal layer of the stomach, resulting in higher satiety and longer gastric emptying times.

In addition, in terms of the baseline characteristics, the IGBI group was significantly younger, mainly because the younger group was more able to accept the riskier endoscopic weight loss method during the consultation period. Because of age and personal lifestyle, blood lipids such as TG, LDL and total cholesterol were all higher in the placebo group. Overall, however, there were no significant differences in BMI, weight, and distribution of overweight or obesity between the two groups, explaining no differences in initial weight levels between the two groups.

In terms of IGBI adverse events, constipation, abdominal pain, and nausea were mostly sporadic, and serious complications were rare. This result is similar to previous journals [40].

The strengths of our research include the following: (1) data were taken from the first real-world Asian weight loss clinic to implement IGBI; (2) the presence of a low-calorie, high-protein diet as a control group increased the comparability of the experimental data. However, there were still limitations to this study, as follows: (1) the use of botulinum in the stomach is still not approved by the U.S. Food and Drug Administration; (2) this was a non-RCT study, so there are still doubts about selection bias, etc.; (3) the follow-up time was not long enough, with only 12 weeks currently analyzed. As far as these clinical data are concerned, the weight loss effect of IGBI is still very promising, and it is worthy of confirmation in larger studies.

## 6. Conclusions

IGBI is a new method that shows that following dietary suggestions from a dietitian can be more effective than diet control alone. Additionally, because of few post-IGBI adverse events, the combination of a low-calorie high-protein diet and IGBI may provide a treatment option for overweight or obese adults in the future, but larger studies are needed to confirm this.

## Figures and Tables

**Figure 1 jcm-11-03325-f001:**
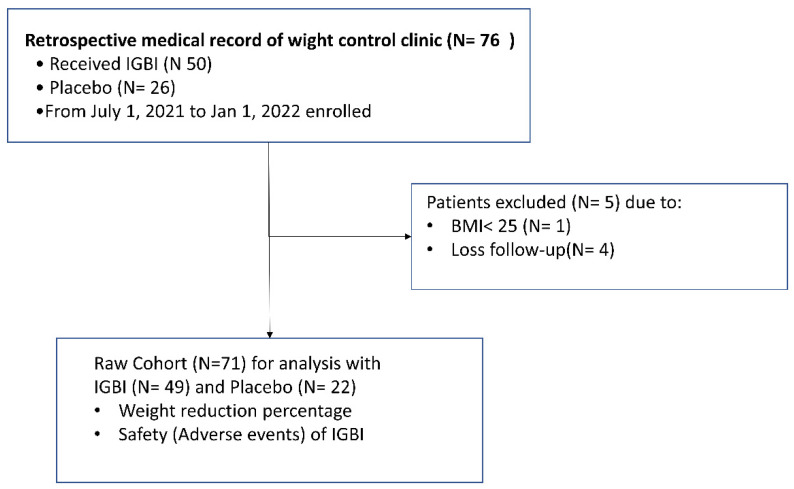
Study flowchart.

**Figure 2 jcm-11-03325-f002:**
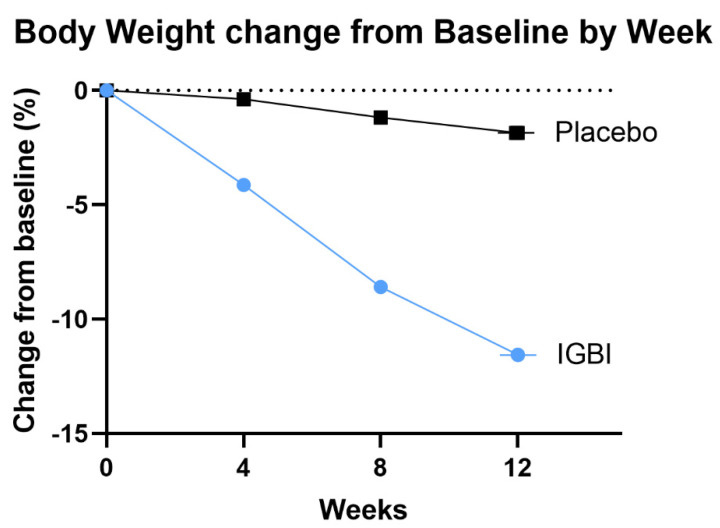
Comparing 12-week body weight change from baseline in the IGBI and placebo groups. IGBI: intragastric botulinum injection.

**Figure 3 jcm-11-03325-f003:**
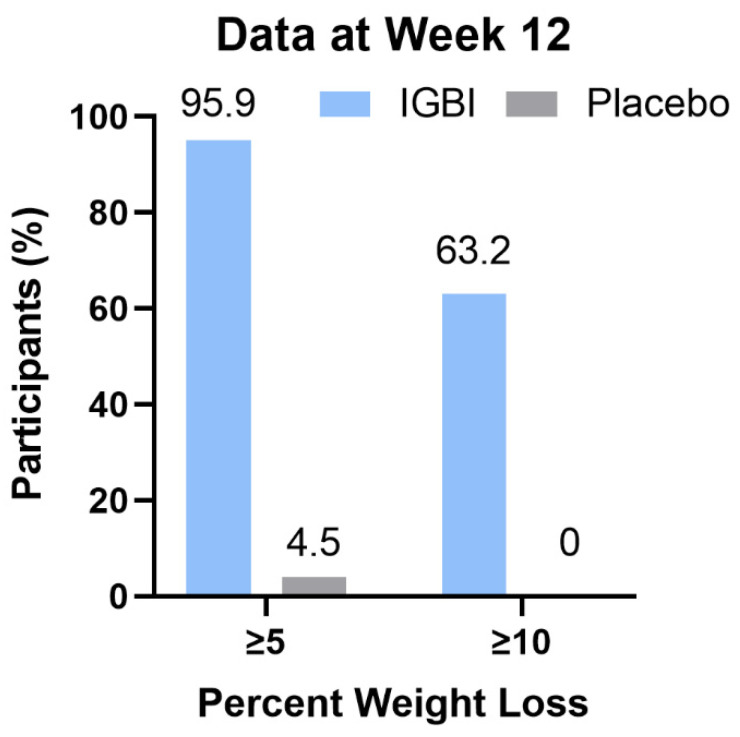
Observed percentages of participants with a body weight reduction of at least 5% and 10% from baseline to week 12 during the study period in the IGBI and placebo groups. IGBI: intragastric botulinum injection.

**Table 1 jcm-11-03325-t001:** Baseline clinical characteristics of the patients.

Characteristic	IGBI (N = 49)	Placebo (N = 22)	*p*-Value
Age (years)	39.8 (37.1–42.4)	49.7 (43.1–56.3)	0.007
Female sex, *n* (%)	37 (76)	16 (72)	0.80
Weight (kg)	82.1 (77.2–87.1)	77.3 (70.7–83.8)	0.25
BMI (kg/m^2^)	29.3 (26.2–33.3)	28.0 (26.8–30.8)	0.63
Overweight, *n* (%)	27 (55%)	12 (54%)	0.66
Obesity, *n* (%)	22 (45%)	10 (46%)	0.66
Fatty liver stage, *n* (%)			0.25
Mild	9 (18%)	3 (13%)	
Moderate	9 (18%)	5 (22%)	
Severe	27 (55%)	14 (63.6%)	
HBA1c (%)	5.5 (5.4–5.6)	5.5 (5.3–5.7)	0.49
Cholesterol (mg/dL)			
Total	164.1 (155.4–172.7)	186.0 (164.1–207.9)	0.005
HDL cholesterol	43.5 (37.6–49.4)	51.8 (43.2–60.5)	0.11
LDL cholesterol	100.1 (92.7–107.5)	134.2 (113.8–154.5)	0.003
Triglycerides (mg/dL)	112.9 (82.8–143.0)	158.6 (132.0–185.2)	0.02

IGBI: intragastric botulinum injection; BMI: body mass index; HBA1c: glycated hemoglobin; HDL: high-density lipoprotein; LDL: low-density lipoprotein. Data are expressed as *n* (%) for categorical data and as mean ± standard deviation or median (interquartile range) for continuous data.

**Table 2 jcm-11-03325-t002:** Adverse events.

Adverse Event	IGBI (49)	Placebo (22)
Time to any adverse event, weeks	1 (1–2)	1 (1)
Any adverse event, no (%)	20 (40.8)	2
Serious adverse events, no (%)	0(0)	0
Reported adverse evet, no. (%)		
Nausea	2 (4.0)	0
Vomiting	0(0)	0
Diarrhea	1 (2.0)	0
Constipation	12 (24.4)	2 (9.0)
Nasopharyngitis	2 (4.0)	0
Urinary disorder	1 (2.0)	0
Abdominal pain	1 (2.0)	0

IGBI: intragastric botulinum injection.

## Data Availability

Data available on request due to restrictions of privacy or ethical.

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
