# Peer review of "Effect of Intragastric Botulinum Type A Injection Combined with a Low-Calorie High-Protein Diet in Adults with Overweight or Obesity"

_jcm, 2022, doi:10.3390/jcm11123325_

Round 1

Reviewer 1 Report

This paper is considered to be important data.

1) Please provide a definition of overweight and obesity.

2) Please describe the low-calorie high-protein diet course.

3) Please describe the diagnostic criteria for Fatty liver stage.

4) In table 1, NAFLD in footnotes but not included in table.

5) Please also list adverse events in control group. If there are no data, this should be mentioned in the limitation.

6) Please also provide age- and sex-adjusted analyses; the analyses in Figure 3 can be adjusted for age and sex using logistic regression analysis.

7) Why were IGIB not performed on patients in the control group?

8) In the discussion, why did the authors state that the sample size was insufficient? The sample size seemed sufficient because both the primary and secondary endpoints were significantly different. If additional analyses could not be performed, please describe what kind of additional analyses should have been performed.

Author Response

Dear reviewer 1:

Thank you for your patience in reviewing our articles, the team appreciates it.

Below is our reply and thank you for your suggestion.

1)Please provide a definition of overweight and obesity.

Response:

Thank you for your valuable advice, it was our negligence that we added ".... For adults, the WHO definition of obesity includes 1. Overweight as body mass index (BMI) greater than or equal to 25. 2 . Obesity as BMI greater than or equal to 30 ..." WHO definition cited, on Line 52-54, at page 2.

2) Please describe the low-calorie high-protein diet course.

Response:

Thanks for your suggestion, we have added "...Low-calorie high-protein diet control program" in the Materials and methods section added as follows:

“At the beginning, we will first understand the dietary status and habits of the participants and record them. When entering the program, we will give a recommended assessment of the dietary composition of the total calories, about 40% from carbohydrates, 30% from protein, and finally about 30% of fat. Finally, we will ask the participants to send back their diet photos through the communication software to calculate the calorie and nutrient composition and give the participants dietary adjustment suggestions.”

. On line 108-114, at page 7.

3) Please describe the diagnostic criteria for Fatty liver stage.

Response:

This was an oversight on our part, and we added the measurement of fatty liver in the Methods paragraph, citing the measurement from the European Association for the Study of the Liver guidelines. On Line 115-121, page 7-8.

4) In table 1, NAFLD in footnotes but not included in table.
Response:

Thank you very much for your reminder, we mistakenly planted NAFLD in the footnote and we deleted it. Thanks again.

5) Please also list adverse events in control group. If there are no data, this should be mentioned in the limitation.

Response:

This is also our negligence, thank you for your guidance. We have added the adverse events of the placebo in table 2, and added "... In addition, in the placebo group, the average time of adverse events is one week, only two people are mainly constipated. . .." in the paragraph of the safety assessment to make the article more complete. On line 174-175, page 11.

6) Please also provide age- and sex-adjusted analyses; the analyses in Figure 3 can be adjusted for age and sex using logistic regression analysis.

Response:

Thanks for your suggestion, we add on the supplementary Table 2 for further for age and sex adjusted using logistic regression analysis as follows:

Supplementary Table 1 – Multivariate logistic regression analysis of factors associated with weight loss more than 5% in IGBI group, adjusted for sex and age

Variable

IGBI

Adjusted OR (95%CI)

P-value

Weight (kg)

1.0(0.95-1.10)

0.431

BMI (kg/m2)

0.9(0.69-1.22)

0.588

HBA1c (%)

0.6(0.11-3.55)

0.604

Cholesterol (mg/dL)

    Total

1.0(0.96-1.17)

0.237

    HDL cholesterol

0.9(0.82-1.01)

0.089

    LDL cholesterol

0.9(0.86-1.05)

0.345

Triglycerides (mg/dL)

0.9(0.98-1.01)

0.550

OR: Odds ratio; 95%CI: 95% confidence interval; IGBI: Intragastric botulinum injection

But the placebo group could not run logistic regression because too few people (1 person) lost 5% of their body weight. Predictors could not be determined.

Thank you for your very good suggestion, we put it in supplementary.

7) Why were IGIB not performed on patients in the control group?
Response:

Thanks for your question.

Because this is a retrospective study, 22 people reject the method of gastroscope and drug weight loss, so they only use the weight loss method recommended by nutritious diet. Based on the participant's right to freely choose a weight loss method in a weight loss clinic. Because of this, it is best to use it as a control group.

8) In the discussion, why did the authors state that the sample size was insufficient? The sample size seemed sufficient because both the primary and secondary endpoints were significantly different. If additional analyses could not be performed, please describe what kind of additional analyses should have been performed.

Response:

Thank you for your valuable suggestion. Indeed, the sample size in our article is sufficient, so it is not a limitation. We removed this sentence.

I think further research is to see if different weight loss methods achieve better results.

For example, IGBI plus semaglutide (GLP-1) is more effective for weight loss when combined with a low-calorie, high-protein diet.

This is our humble opinion, but there is still a long way to go to lose weight, and we hope to provide a little help to medical treatment in the future.

Reviewer 2 Report

Title:  Effect of Intragastric Botulinum Type A Injection combined with a low-calorie high-protein diet in adults with overweight or obesity

Authors: Po-Ke Hsu, Chia-Lin Wu, Yu-Hsuan Yang, and James Cheng-Chung Wei

General Comment:

In the world struggling with the obesity pandemic, there is a constant need for novel therapeutic options. In their work, Po-Ke Hsu et al. evaluated the potential of intragastric botulinum type A injection (IGBI) in treating overweight and obesity. They found that combining a low-calorie, high-protein diet and IGBI is an effective and safe procedure for weight reduction in overweight or obese patients. The study concept is interesting, and the results may have clinical implications; however, the authors should address some issues before it is accepted for publication.

Major revisions:

Abstract

Please add data on patients' BMI to make the abstract more informative.

Introduction

The introduction is relatively brief. Please consider providing a broader background to the study.

Material and methods

Please provide information on how liver steatosis was assessed in the study protocol.

Discussion

Discussion is also relatively brief; please consider discussing the results in the light of, e.g., animal studies and already published clinical trials. It would also be valuable to suggest further directions in this area of research.

Moreover, the author writes: “In addition, in the baseline characteristic, the IGBI group was significantly older, mainly because the younger group was more able to accept the riskier endoscopic weight-loss method during the consultation period.”

The mean age in the IGBI group was 39.8 years, while in the Placebo – 49.7 years, so the IGBI group was younger.

Minor revisions:

Whole manuscript:

-          please improve manuscript editing;

-          the manuscript may benefit from the assistance of a native English speaker

Introduction

“However, there are many studies on the weight loss effect of IGBI is still controversial [11].” – please correct this sentence from the grammatical point of view and provide some more references.

Material and methods

Subsection Endoscopic Procedures

This section is written rather carelessly – with a combination of different tenses, part of it seems to be "copy-pasted" from the study protocol – please consider re-editing this part of the manuscript.

Results

“In terms of overweight and obesity, the (percentage) was 27(55%), 22(45%) in IGBI group and 12(54%),10(46%) in the placebo group, respectively without significant, p= 0.66.“ the word “difference” seems to be missing.

Author Response

Dear reviewer 1:

Thank you for your patience in reviewing our articles, the team appreciates it.

Below is our reply, and thank you for your suggestion.

Major revisions:

Reviewer 2 comment:

  1. Abstract

Please add data on patients' BMI to make the abstract more informative.

Response:

Thank you for your valuable suggestion, we have added the BMI information in the method and result paragraph of the abstract as follows:

In method paragraph of abstract:”… We retrospectively collected medical data from July 1, 2021 to January 1, 2022, with a total of 71 patients without diabetes who participated in the bariatric clinic with body mass index (BMI) > 25kg/m2….”.

In result paragraph of abstract: “…In the characteristic of two groups, the mean bmi was 29.3 kg/m2 in IGBI group, and 28.0 kg/m in placebo group, p=0.63 without significant difference.”

Thanks again for your suggestion, I think this abstract will be more specific and clearer.

Reviewer 2 comment:

  1. Introduction

The introduction is relatively brief. Please consider providing a broader background to the study.

Response:

Thank you for your valuable advice.

We added in introduction:

  1. The WHO epidemiological description is as follows: "...According to the World Health Organization (WHO) in 2014, approximately 1.9 billion adults (over the age of 18) are overweight and more than 600 million are obese [1]. By 2030 , the number of overweight people is expected to exceed 2.16 billion and the number of obese people will exceed 1.12 billion [2]. For adults, the WHO definition of obesity includes 1. Overweight as body mass index (BMI) greater than or equal to 25. 2. Obesity as BMI greater than or equal to 30 ..." On Line 51-55, page 5.
  2. Recommendations of the AHA guideline" ...In the AHA obesity guidelines, all patients recommended for weight loss should be offered or referred for a comprehensive lifestyle intervention [8]. Comprehensive lifestyle intervention with individualized calorie-restricted diet such as dietary intake of 1,200- 1,500 kcal/d for women and 1,500-1,800 kcal/d for men is recommended. Dietary control, such as a low-calorie, high-protein diet, is a good choice to weight loss [9]. ... " On Line 59-64, page 5.

Reviewer 2 comment:

  1. Material and methods

Please provide information on how liver steatosis was assessed in the study protocol.

Response:

This is really our negligence. We will add the measurement method of fatty liver and related literature to the material and method as follows:

In material and method paragraph

“Definition and assessment of fatty liver. We use ultrasound B mode to evaluate indicators of fatty liver such as: liver brightness, contrast between the liver and the kidney, US appearance of the intrahepatic vessels. If there is no match, score: 0 means no fatty liver. Mild fatty liver is a score of 1, which means that the liver is bright and other indicators are normal. Moderate fatty liver refers to two points, which means liver bright and contrast between the liver and the kidney. Severe fatty liver refers to 3 points, the liver echogenicity is significantly enhanced, and the portal vein wall imaging is poor[17].” On line 104-110, Page 7.

Reviewer 2 comment:

  1. Discussion

Discussion is also relatively brief; please consider discussing the results in the light of, e.g., animal studies and already published clinical trials. It would also be valuable to suggest further directions in this area of research.

Response:

Thank you for your valuable suggestion, we have added information about In vivo and in vitro animal studies. In addition, the first case report and pilot study on the treatment of obesity with botulinum type A are also added as follows in Discussion section:

“…In vivo and in vitro studies in 2003 and 2004 explored the effects of botulinum type A and gut sympathetic nerves [27] [28]. In addition, James et al also reported in vitro animal study that botulinum type A can effectively reduce the contraction of antrum muscle [29]. Gui et al demonstrated that injection of botulinum toxin A into antrum was effective in reducing food intake by 37.8% and weight loss by 14% in rats, and the effect persisted for up to 2 months [30].

In 2003, Rollnik et al. documented the first case report of antral Injections of botulinum A toxin for the Treatment of Obesity and showed that 4 weeks after the injection, the weight reduced from 100.6 kg to 91 kg, and the BMI decreased from 31 kg/m2 to 29 kg/m2 [31]. Later, a pilot study also showed that botulinum toxin antral injection can delay gastric emptying time and cause a decrease in appetite, thereby allowing obese patients to lose weight [32]….” On line 199-208, page 13.

Thank you for the reminder to enrich our discussion paragraph.

Reviewer 2 comment:

  1. Moreover, the author writes: “In addition, in the baseline characteristic, the IGBI group was significantly older, mainly because the younger group was more able to accept the riskier endoscopic weight-loss method during the consultation period.”

The mean age in the IGBI group was 39.8 years, while in the Placebo – 49.7 years, so the IGBI group was younger.

Response:

Thank you for your reminder, this is a major mistake in our writing. We changed "older" to "younger". On line 221, page 14.

Minor revisions:

Reviewer 2 comment:

  1. Whole manuscript:

please improve manuscript editing;

the manuscript may benefit from the assistance of a native English speaker

Response:

Thank you very much for your suggestion, we look for the assistance of a native English speaker for English editing.

Reviewer 2 comment:

  1. Introduction

“However, there are many studies on the weight loss effect of IGBI is still controversial [11].” – please correct this sentence from the grammatical point of view and provide some more references.

Response:

Thank you for your patient review.

This paragraph actually expresses that due to the lack of real-world IGBI data, there is still insufficient evidence for IGBI for sustained weight loss.

We changed it to: "However, current real-world data are insufficient to show that IGBI is an effective method for sustained weight loss in obese patients." As stated in the reference, current evidence is still insufficient. On Line 71-72, Page 5.

Reviewer 2 comment:

  1. Material and methods

Subsection Endoscopic Procedures

This section is written rather carelessly – with a combination of different tenses, part of it seems to be "copy-pasted" from the study protocol – please consider re-editing this part of the manuscript.

Response:

This is also our major oversight, and we re-wrote the entire endoscopic procedure to make the content clearer.

We modify it as follows:

“ First, we specified and signed an informed consent form for each participant who chose IGBI. And arrange blood tests and physical examinations to ensure that there are no related risks. If all inspection data are satisfactory, we will schedule a time for IGBI. IGBI is performed by qualified and licensed specialists certified by the Digestive Endoscopy Society of Taiwan. All participants underwent esophagogastroduodenoscopy under fentanyl and propofol sedation. IGBI is not performed if there is gastric ulcer, tumor, or food residue during gastroscopy. During the procedure,  we mix 30ml of 0.9% normal saline with botulinum type A (Botox; Allergan, Irvine, CA) containing 300 units through a 25-gauge needle, and inject 10ml each on antrum, body, and fundus through submucosal injection. After the procedure, we use ultrasound to confirm whether there is ascites or free-air to ensure there are no complications such as gastric perforation and bleeding.  Participants then returned weekly for physical exam records and dietary recommendations… “ on line 89-107, page 6-7.

Reviewer 2 comment:

  1. Results

“In terms of overweight and obesity, the (percentage) was 27(55%), 22(45%) in IGBI group and 12(54%),10(46%) in the placebo group, respectively without significant, p= 0.66.“ the word “difference” seems to be missing.

Response:

Thank you for your guidance, this is also an obvious error in our text.

We will add " difference " to become "In terms of overweight and obesity, the number (percentage) was 27(55%), 22(45%) in IGBI group and 12(54%),10(46%) in placebo group, respectively without significant difference, p= 0.66." On line 156, page 9.

Round 2

Reviewer 2 Report

I want to express my gratitude for the opportunity to re-review the paper entitled: " Effect of Intragastric Botulinum Type A Injection combined with a low-calorie high-protein diet in adults with overweight or obesity" by Po-Ke Hsu et al. Since the authors addressed my concerns regarding the methodology and significantly improved the Introduction and Discussion sections, I find the manuscript acceptable for publication in Journal of Clinical Medicine.